# Effect of Omega-3 Rich High-Fat Diet on Markers of Tissue Lipid Metabolism in Glucocorticoid-Treated Mice

**DOI:** 10.3390/ijms241411492

**Published:** 2023-07-15

**Authors:** Wangkuk Son, Katie Brown, Aaron Persinger, Ashley Pryke, Jason Lin, Zereque Powell, Noah Wallace, Marie van der Merwe, Melissa Puppa

**Affiliations:** College of Health Sciences, University of Memphis, Memphis, TN 38152, USA

**Keywords:** lipid metabolism, liver metabolism, muscle lipid metabolism, dexamethasone, polyunsaturated fatty acids, PUFAs

## Abstract

Glucocorticoids (GCs) are some of the most widely prescribed therapies for treating numerous inflammatory diseases and multiple cancer types. With chronic use, GCs’ therapeutic benefits are concurrent with deleterious metabolic side effects, which worsen when combined with a high-fat diet (HFD). One characteristic of the common Western HFD is the presence of high omega-6 polyunsaturated fatty acids (PUFAs) and a deficiency in omega-3 PUFAs. The aim of this experiment was to determine whether fat composition resulting from HFD affects glucocorticoid-induced alterations in lipid-handling by the liver and skeletal muscle. Male wild-type C57BL/6 mice were randomized into two groups: n-6 (45% fat 177.5 g lard) and n-3 (45% fat 177.5 g Menhaden oil). After 4 weeks on their diets, groups were divided to receive either daily injections of dexamethasone (3 mg/kg/day) or sterile PBS for 1 week while continuing diets. The n-3 HFD diet attenuated adipose and hepatic fatty accumulation and prevented GC-induced increases in liver lipid metabolism markers *Cd36* and *Fabp*. N-3 HFD had little effect on markers of lipid metabolism in oxidative and glycolytic skeletal muscle and was unable to attenuate GC-induced gene expression in the muscle. The present study’s result demonstrated that the change of fat composition in HFD could beneficially alter the fatty acid accumulation and associated lipid metabolism markers in mice treated with dexamethasone.

## 1. Introduction

In recent decades, the prevalence of the Western diet has grown and dramatically increased obesity, which is a serious and growing health problem in the world [1]. The present “Western” diet or high-fat diet (HFD) is characterized by high levels of n-6 polyunsaturated fatty acid (PUFA) and a deficiency in n-3 PUFA (20:1 n-6/n-3), while the ideal ratio of n-6/n-3 is 2:1 [2]. Since signaling molecules generated from n-6 PUFA exhibit pro-inflammatory properties [3], the accumulation of n-6 PUFAs can lead to the lipid metabolism disorder and metabolic syndrome [4], building the foundation of various diseases such as coronary heart disease, stroke, diabetes, and some types of cancer [5,6]. Therefore, a better understanding of the types of dietary fat consumed and their influence on whole-body metabolism is necessary.

In the postprandial state, once lipids are used, the rest are stored mainly in white adipose tissue (WAT) and the liver in the form of triglycerides (TGs) [7,8]. In chronic Western diet conditions, TG accumulation exceeds storage limitations, which leads to hypertrophy and dysfunction in WAT and liver cells due in part to the high levels of n-6 PUFA in the Western diet [7,8]. Inflammation caused by excessive TG accumulation and HFDs rich in n-6 PUFA are involved in various cellular dysfunctions related to macrophage infiltration, mitochondrial oxidation, and transcription changes [9]. These changes disturb insulin and glucose uptake, which accelerates lipolysis and fatty acid spillage into circulation and build the foundation for the development of metabolic syndromes [10].

Glucocorticoids (GCs) are a steroid hormone and some of the most widely prescribed therapies for treating numerous inflammatory diseases and multiple cancer types [11]. GCs regulate various physiologic processes for life and exert anti-inflammatory and immunosuppressive actions [12]. However, in chronic conditions with high dose treatments, GCs’ therapeutic benefits are concurrent with the deleterious side effects including lipid metabolism disruption and metabolic syndrome [13]. In addition, the lipid metabolism disruption induced by GCs becomes worse when combined with a HFD. For instance, Anna et al. found that a combination of a HFD with GCs resulted in a greater increased hepatic lipid, collagen content, and hepatic steatosis compared to GCs or HFD alone [14]. Chronic GC use contributes to the development of lipid metabolism dysfunction and metabolic syndrome factors, which resemble the phenotype associated with HFD feeding, and the effects are amplified when GCs are combined with HFD.

In contrast to n-6 PUFA, the signaling molecules generated by n-3 PUFA exhibit anti-inflammatory properties, improving lipid metabolism, adiposity, inflammatory cytokines, and insulin–glucose homeostasis [15]. Therefore, n-3 PUFA has become one of the dietary interventions studied actively to improve lipid metabolism. A recent study by Sakamuri et al. demonstrated that a low n-6: n-3 PUFA ratio (2:1) prevented high-fructose-induced dyslipidemia, hepatic oxidative stress, and inflammation with the reduction of adiposity and circulatory triglycerides [16]. Additionally, Hill et al. found that mice fed a high n-3 HFD with GCs showed decreased epididymal white adipose tissue (eWAT) weight, adipose size, and glucose concentration compared to the mice fed a high n-6 HFD with GCs [17]. In light of these findings, it is becoming clear that high n-3 PUFA intervention can positively affect lipid metabolism dysfunction induced by GCs and HFD.

Hypertrophied WAT and fatty liver induced by the chronic HFD state release excessive lipids into circulation [18]. The released lipid flux is deposited into other organs, such as skeletal muscle, that are not well-equipped to store lipids, causing lipotoxic stress and cell dysfunction [18]. The deposited fatty acid in skeletal muscle activates serine/threonine kinases that impair insulin receptors’ ability to stimulate downstream pathways, decreasing the translocation of glucose transporter type 4 (GLUT4) and, therefore, reducing glucose uptake into skeletal muscle cells [19]. Skeletal muscle is involved in the clearance of 25% of circulating glucose in a basal fasting state and up to 70–85% in a postprandial state [20]. Therefore, increased lipid deposition in skeletal muscle can be the main contributor to hyperglycemia in circulation [21], promoting lipid metabolism dysfunction in adipose tissue and the liver [13]. However, the research on HFD composition and lipid metabolism induced by HFD with GCs in skeletal muscle is limited. Therefore, the purpose of this study was to determine whether fat composition in an HFD affects GC-induced alterations in lipid handling by the liver and skeletal muscle.

## 2. Results

### 2.1. Body Weight and Body Composition

Body weight was recorded throughout the duration of the study. The main effect of time on body weight (*p* < 0.001) was expected as the mice grew (Figure 1A). During the last three days of dexamethasone injection, the mice on the n-6 diet receiving dexamethasone weighed significantly more than the mice on the n-3 diet receiving dexamethasone; however, this difference was not seen after fasting on the day of euthanasia (Figure 1A).

Body composition was measured at the beginning of week one before the mice started on the n-6 and n-3 rich diets, at the end of week 4 after consuming the diets for 4 weeks, and at the end of week 5 after receiving either PBS or dexamethasone injections daily for one week. Lean body mass increased throughout the duration of the study (*p* < 0.001). After one week of dexamethasone treatment, the main effect of dexamethasone was to decrease lean body mass (*p* = 0.04). Lean body mass was unaffected by diet (Figure 1B).

Total body fat composition increased with time (*p* < 0.0001). At week 4, the main effect of n-6 was to increase body fat (*p* = 0.04). After 1 week of dexamethasone treatment, mice fed the n-6 diet receiving dexamethasone had increased body fat, while the mice consuming the n-3 diet were protected from increased body fat after 1 week of dexamethasone treatment (Figure 1C).

To further delineate changes in body composition, several tissues were harvested and weighed at the time of sacrifice. There was no effect from n-3 on hindlimb muscle mass; however, there was an effect from dexamethasone in reducing total hindlimb muscle mass (Figure 2A). Spleen mass increased with the n-3 rich diet, and dexamethasone decreased spleen mass regardless of diet (Figure 2B). Epididymal fat mass decreased in n-3-fed mice regardless of dexamethasone treatment (Figure 2C,D). Finally, there was an effect of dexamethasone to increase liver mass in n-6 HFD fed mice, while liver mass was not increased in the n-3 HFD fed mice treated with dexamethasone (Figure 2E).

### 2.2. Liver Lipid Regulation

We next looked at the liver to better understand the increased mass that was induced via dexamethasone treatment. Upon histological analysis, n-6 HFD fed mice treated with dexamethasone displayed an accumulation of lipids in the liver. This accumulation was not seen in the n-3 HFD fed mice that received dexamethasone (Figure 3A). Triglyceride analysis of the liver tissue revealed that one effect of the n-6 HFD was to increase liver triglyceride content (*p* = 0.01); however, no effect of dexamethasone was seen (Figure 3B). Because lipid infiltration is often associated with inflammation and an increase in spleen size was noted, we measured several markers in macrophage infiltration and inflammation in the liver. While neither dexamethasone nor diet had any effect on *Cd68* or *Il-6* mRNA levels, one effect of the omega-3 diet was lowering *Itgam* mRNA expression (Figure 3C).

To further understand the impact of omega-3 and omega-6 high-fat diets on liver lipid accumulation during dexamethasone treatment, we analyzed the liver for transcriptional markers of lipid metabolism. Carnitine acetyltransferase (Crat) regulates the ratio of acyl-CoA to CoA and is an important mediator for the delivery of fatty acids for oxidation. *Crat* mRNA increased with the n-3 rich diet (*p* = 0.01); however, there was no effect of dexamethasone on *Crat* (Figure 4A). One week of dexamethasone treatment had no effect on hormone sensitive lipase (Lipe), but there was an overall increase in liver *Lipe* expression with the n-3 rich diet (*p* = 0.04) (Figure 4B). Fatty acid-binding protein 3 (*Fabp3*) expression increased in the n-6 fed mice treated with dexamethasone, while *Fabp2* decreased in n-6-fed mice given dexamethasone. The increase in *Fabp3* expression was completely ablated in n-3-fed mice. Neither diet nor dexamethasone had any effect on *Fapb1* expression (Figure 4C). CD36 facilitates fatty acid transport across the cell membrane. Similar to the effects seen in *Fabp*, dexamethasone increased *Cd36* expression in the livers of mice receiving the n-6-rich diet; however, an n-3-rich diet completely attenuated the dexamethasone-induced increase in *Cd36* mRNA (Figure 4D). There was no effect of either diet or dexamethasone on expression of fatty acid synthase (*Fasn*) (Figure 4E), carnitine palmitoyltransferase 1A (*Cpt1a*), a target of par-alpha activity (Figure 4F), or liver x receptor (*Lxr*) (Figure 4G). Taken together, these data suggest that the increased lipids seen in the livers of n-6 HFD plus dexamethasone-treated mice are in part due to increased *Cd36* and *Fabp3* expression, which is prevented with n-3 HFD feeding.

### 2.3. Muscle LIPID Regulation

Next, we examined the expression of lipid metabolism regulating transcripts in skeletal muscle. Skeletal muscle is a large regulator of both carbohydrate and lipid metabolism. Dysregulation of skeletal muscle lipid metabolism has also been implicated in the pathology of metabolic disorders such as insulin resistance, metabolic syndrome, and cancer cachexia [22,23,24]. In glycolytic skeletal muscle, there was no effect from diet or dexamethasone on *Crat* expression (Figure 5A). Although not significant, there was a trend for dexamethasone to increase *Lipe* expression in the glycolytic portion of the gastrocnemius muscle (*p* = 0.054) (Figure 5B). N-3-rich high fat diets increased expression of *Fabp* in glycolytic muscle (*p* = 0.04) (Figure 5C); however, there was no effect from either dexamethasone or diet on *Cd36* (Figure 5D) and *Fasn* (Figure 5E) expression in glycolytic muscle.

Oxidative muscle has a higher capacity to metabolize fats than glycolytic muscle. In the oxidative portion of the gastrocnemius muscle, dexamethasone had the effect of increasing *Crat* (Figure 6A), *Lipe*, (Figure 6B) and *Fabp* (Figure 6C) expression, which were unaffected by diet. Similar to the glycolytic muscle portion, there was no effect from either diet or dexamethasone on *Cd36* (Figure 6D) and *Fasn* (Figure 6E) expression in the oxidative muscle. These data demonstrate a differential regulation of these fatty acid metabolism in the liver and skeletal muscle during dexamethasone treatment, with the liver demonstrating more robust responses to diet and dexamethasone treatment.

## 3. Discussion

In the present study, we aimed to determine whether the fat composition of HFD affects GC-induced alteration in lipid-handling by the liver and skeletal muscle. We observed that omega-3 HFD improved adipose tissue mass and liver fat accumulation and the associated lipid metabolism markers. We also reported that one week of dexamethasone treatment exacerbates the lipid condition induced by an omega-6 HFD. Feeding mice with diets rich in omega-3 protected mice from adipose tissue hypertrophy and hepatic steatosis by decreasing fat accumulation. Further, omega-3 HFD also reversed the increased lipid transport markers (CD36 and FABP3) in the liver, which play an essential role in fatty acid inflow.

Considering that body weight change has been confirmed in longer experiments such as over five weeks [13,14,16,25,26], the short-term lipid dysfunction induced by HFD and GCs in our study do not seem to have reached that point. Moreover, since the side effects of GC treatment involve skeletal muscle loss, we need to consider the body weight complementary action between fat increase and muscle loss induced. As expected, the omega-6 diet with dexamethasone treatment significantly increased fat mass, which indicates synergetic effects. Chronic omega-6 HFD diet leads to fatty acid over-accumulation in adipose tissue and liver-activating fibrosis, hypoxia, and inflammation pathways. The cells’ storage ability becomes dysfunctional, finally contributing to unhealthy fat mass increase and insulin resistance [8].

Previous studies have demonstrated that omega-3 HFD diets improve fat mass by stabilizing lipid metabolism due to their anti-inflammatory action [25,27]. Likewise, in our study, the omega-3 HFD diet maintained fat mass even with dexamethasone treatment. This data shows that fat composition in HFD is essential to ameliorating GC-induced fat mass increase. In the present study, an omega-3 HFD diet showed a lower eWAT weight than omega-6 diet groups and relatively smaller adipocytes, which indicates that omega-3 PUFAs may protect white adipose tissue from becoming hypertrophied. It is possible that omega-3 helps white adipose tissue store excessive fatty acid via differentiation of resident tissue precursors and creating new adipocytes; however, more research is needed. Simultaneously, omega-6 and dexamethasone treatment encourage white adipose tissue to store excessive fatty acid via enlargement of existing adipocytes [8,28]. Combined with cellular dysfunction, this process would be associated with insufficient vascularization, hypoxic response, tissue fibrosis, and inflammation.

Similar to the eWAT, an omega-3 HFD diet showed little change to liver weight and triglyceride content with treatment with dexamethasone, while omega-6 and dexamethasone treatment increased both size and triglyceride content in the liver. The liver is a critical regulator of lipid metabolism and whole-body energy homeostasis. In the setting of chronic HFD, the liver conveys fatty acids via de novo lipogenesis to adipocytes, which are responsible for 20% of newly accumulated triglycerides in adipose tissue [29] and in liver tissue [7]. Because of this, the liver is sensitive to lipid over-accumulation and the damage induced by lipid metabolism dysfunction. In the present study, the omega-3 HFD diet inhibited the increase of liver fat accumulation. Additionally, the omega-3 HFD decreased the expression of itgam, a marker of macrophages without changes in liver Il-6 expression. The omega-3 effect may be partially explained by involving hepatic antioxidant enzyme, lipogenic, inflammatory, and oxidative ER stress gene expression as well as pre-receptor amplification of GCs as demonstrated by Sakamuri et al. [16].

The present data showed that an omega-6 HFD with dexamethasone treatment significantly increased *Cd36* and *Fabp3*, which were attenuated by the omega-3 diet. CD36 and FABP are proteins that facilitate FFA transport into various tissues across the plasma membrane by binding saturated and long unsaturated chain fatty acids [30,31]. It is possible that the omega-3 HFD would prevent the liver from receiving excessive fatty acids through the prevention of fatty acid import. Interestingly, the liver-specific FABPs were unaltered by diet; however, the animals only consumed a high-fat diet for 5 weeks, which may not be long enough to see large transcriptional changes. Additionally, *Lipe* and *Crat* were both significantly higher in the omega-3 HFD group compared to omega-6 HFD. Hormone-sensitive lipase is responsible for breaking down triglycerides and releasing fatty acids into circulation, and Crat plays a role in cell energy metabolic flexibility [10,32]. Therefore, hormone-sensitive lipase and Crat’s significant increase in the omega-3 HFD group may indicate improved fatty acid metabolism compared to the omega-6 HDF and may be supported by the beneficial effect of omega-3 on inflammation, oxidative stress, and hepatic antioxidant enzyme function [16]. Future studies should also examine protein levels of fatty acid metabolism regulators as there may be alterations in mRNA stability that would not necessarily reflect the same changes that would be seen at the protein level.

Research is still lacking on the effects of fat composition on lipid metabolism in skeletal muscle, even though skeletal muscle is a large contributor to hyperglycemia, insulin resistance, and lipid metabolism dysfunction. The present data in glycolytic and oxidative muscle showed that some markers (*Fabp* and *Lipe*: glycolytic/*Fabp*, *Lipe* and *Crat*: oxidative) have a significant difference only induced by dexamethasone treatment. Although significant differences are seen in the liver, it is possible that this lipotoxicity does not reach skeletal muscle during short-duration glucocorticoid treatment. On the other hand, dexamethasone treatment has affected a few lipid metabolism markers. Based on previous research results [33,34], GCs can be involved in lipid mobilization and insulin resistance in skeletal muscle. These effects may be regulated by the 11β-hydroxysteroid dehydrogenase type1 (11β-HSD1), which is involved in generating active glucocorticoids in tissues [33,34]. With these results, however, we cannot explain GC’s effect on lipid metabolism in skeletal muscle. Future research should consider the difference of acute and chronic GC effects on skeletal muscle specific to fatty metabolism.

## 4. Materials and Methods

### 4.1. Animals and Experimental Design

All experimental and housing protocols were approved by the Institutional Animal Care and Use Committee of the University of Memphis, protocol number 0830. C57BL/6 male mice, 7 weeks of age, were purchased from Envigo (Indianapolis, IN, USA). All animals were kept on a 12:12-h light–dark cycle and provided ad libitum access to food and water during the study. After 3 days of acclimation, animals underwent baseline testing, including MRI, after being fasted for 5 h. Mice were then randomized into two groups initially to receive either a high-fat diet rich in omega-6 (n-6, 45% fat (177.5 g lard), 35% carbohydrate, and 20% protein, n-6:n-3 PUFA, 13:1) or a high-fat diet rich in omega-3 (n-3, 45% fat (177.5 g Menhaden oil), 35% carbohydrate, and 20% protein, n-6:n-3 PUFA, 1:3, diet composition listed in Table 1). After 4 weeks on their respective diets, both groups were divided, with half of the mice receiving either a subcutaneous injection of dexamethasone (3 mg/kg body weight) or sterile PBS while continuing their current diet throughout the 5th and last week. We closely monitored the daily body weight, food intake, and grooming of all mice for the experiment’s duration. The remaining food (from the previous day’s consumption) and body weight were measured 3 times a week during the first four weeks and daily following dexamethasone injections. MRI testing was performed on mice at 8, 12, and 13 weeks of age to assess body composition (ECO MRI-100, Houston, TX, USA). Blood was collected from all animals at 8, 12, and 13 weeks of age via the facial vein. Approximately 50–100 µL of blood was collected into an EDTA-containing tube for plasma collection. At the end of the study, animals were fasted for 5 h before harvesting tissue. Tissue collection was completed with mice anesthetized with isoflurane and euthanized via cervical dislocation.

### 4.2. Non-Survival Surgery

At the end of the study (14 weeks of age), all animals were fasted for 5 h prior to harvesting tissue. Tissue collection was completed with the mice anesthetized via isoflurane (2–5%). Mice were euthanized by cervical dislocation while anesthetized. Hindlimb skeletal muscles, liver, epididymal fat pad, heart, and spleen were excised. The gastrocnemius muscle was divided into the highly glycolytic white portion and the highly oxidative red portion before being frozen. A portion of liver and white adipose tissue was fixed in 10% neutral buffered formalin, while the rest was and snap frozen in liquid nitrogen for further analysis. Tibias were also removed and measured as a correction factor for body size.

### 4.3. Triglyceride Assay

Triglyceride assay was performed on a serum to determine differences in TG concentration. Free glycerol reagent and the triglyceride reagent (Sigma, St. Louis, MO, USA) were set up according to the manufacturer’s instructions. An initial absorbance of blank, standard, and sample were measured using a spectrophotometer at 540 nm (BioTek, Synergy2, Winooski, VT, USA). Triglyceride reagent was mixed with each sample and incubated at room temperature for 5 min. The second absorbance of blank, standard, and sample was measured at 540 nm. Triglyceride concentration was calculated according to the manufacturer’s instructions.

### 4.4. RNA Isolation and qPCR

The following genes were analyzed for expression: *Cd36*, *Fabp3*, *Lipe*, *Fasn*, and *Crat*. To isolate RNA from mouse liver and white and red gastrocnemius muscle, tissues were homogenized in 3–5 mL Trizol as previously described [35] and quantified using a Nanodrop (ThermoFisher Scientific, Waltham, MA, USA). For qPCR measurement of RNA transcripts, 1 µg of RNA was reverse-transcribed to cDNA. The cDNA was mixed with forward and reverse primers for the intended gene target and SYBR Green qPCR master mix. Gene expression was measured using QuantStudioTM 6 flex system QuantStudio Real-Time PCR software v1.3 (Thermo Fisher Scientific, Waltham, MA, USA). PCR analysis was carried out with using 2^−ΔΔCT^. See Table 2 for primer sequences.

### 4.5. Histology

A representative sample of liver and eWAT was fixed in a 10% neutral buffered formalin solution for 48–72 h. These samples were dehydrated in a series of graded ethanol solutions, cleared with xylene, and then embedded in paraffin. Five-μm sections were stained with hematoxylin and eosin (H&E). Histological analysis was performed using an Imager M7000 microscope (Invitrogen, EVOS, M7000 Imaging system, Waltham, MA, USA). Representative images are presented from each group at 10–20× magnification.

### 4.6. Statistical Analysis

All data are represented as means ± SE. A two-way ANOVA was used to determine the effects of diet and dexamethasone treatment using GraphPad Prism 8. Tukey post hoc analysis was used to examine interactions. Significance was set at *p* ≤ 0.05.

## Figures and Tables

**Figure 1 ijms-24-11492-f001:**
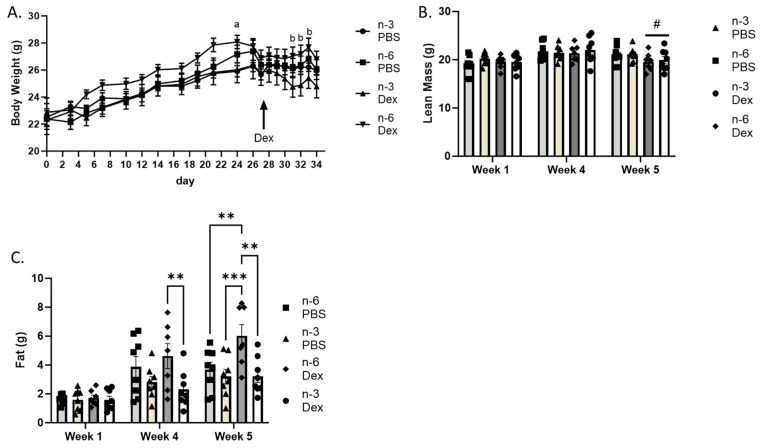
Effects of n-6 or n-3 HFD and dexamethasone on body weight and composition. (**A**) Body weight was measured three times weekly for the first four weeks of the study. On the fifth week, body weight was measured daily prior to dexamethasone injection. MRI was conducted on all mice at baseline at week one, after 4 weeks on either n-3 or n-6 rich diet, and after one week of either PBS or dexamethasone injection, week 5. (**B**) Lean body mass and (**C**) fat mass were recorded. All data are presented as mean ± SEM. Repeated measures via two-way ANOVA were used to analyze changes over time. Significance was set at *p* < 0.05. a signifies difference from n-3 PBS to n-6 Dex at the timepoint, b signifies difference from n-3 Dex to n-6 Dex at the timepoint. # represents a main effect of Dex at the timepoint, ** *p* < 0.01, *** *p* < 0.001.

**Figure 2 ijms-24-11492-f002:**
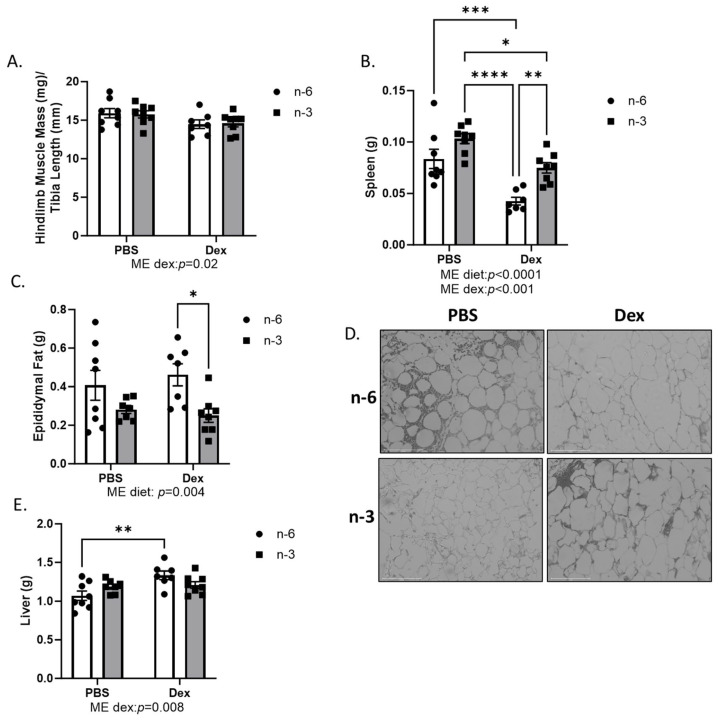
Effects of n-6 or n-3 HFD and dexamethasone on tissue mass. (**A**) Hindlimb skeletal muscle mass was calculated as the sum of the gastrocnemius, soleus, plantaris, quadriceps, tibialis anterior, and extensor digitorum longus muscles after being weighed at the time of euthanasia. (**B**) Spleen mass was measured at the time of euthanasia and recorded. (**C**) Epididymal fat pad was excised and weighed at the time of euthanasia. A portion of the fat pad was fixed in 10% formalin solution for (**D**) representative histological images of the epididymal fat pad (20× magnification), scale bar 150 µm. (**E**) Liver was removed and weighed at the time of euthanasia. All data are presented as mean ± SEM. Two-way ANOVA was used to analyze changes. Significance was set at *p* < 0.05. * *p* < 0.05, ** *p* < 0.01, *** *p* < 0.001, **** *p* < 0.0001. Main effects are denoted below each graph.

**Figure 3 ijms-24-11492-f003:**
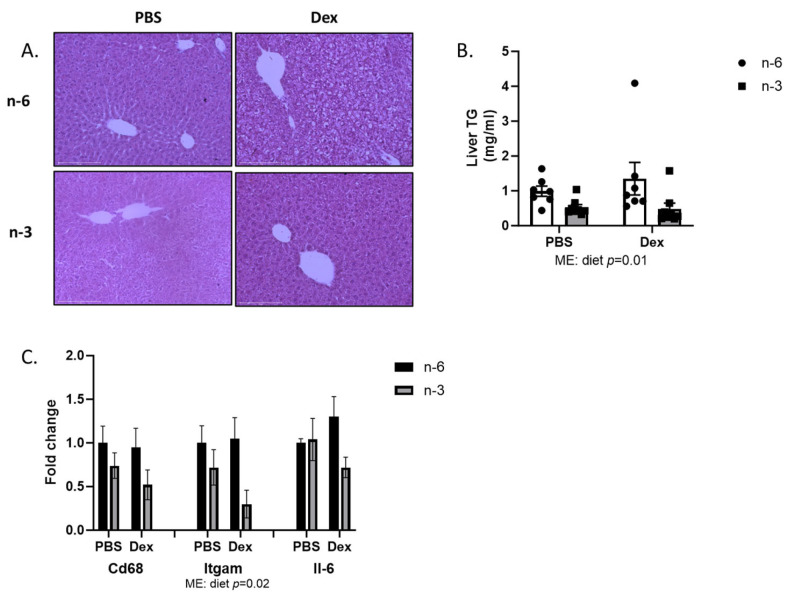
Effects of n-6 or n-3 HFD and dexamethasone on liver lipid accumulation and inflammation. (**A**) Representative H&E images of the liver from each group (10× magnification), scale bar 275 µm. (**B**) Liver triglyceride content was measured. (**C**) mRNA from macrophages and inflammation markers in liver. All data are presented as mean ± SEM. Two-way ANOVA was used to analyze changes. Significance was set at *p* < 0.05. Main effects are denoted below each graph.

**Figure 4 ijms-24-11492-f004:**
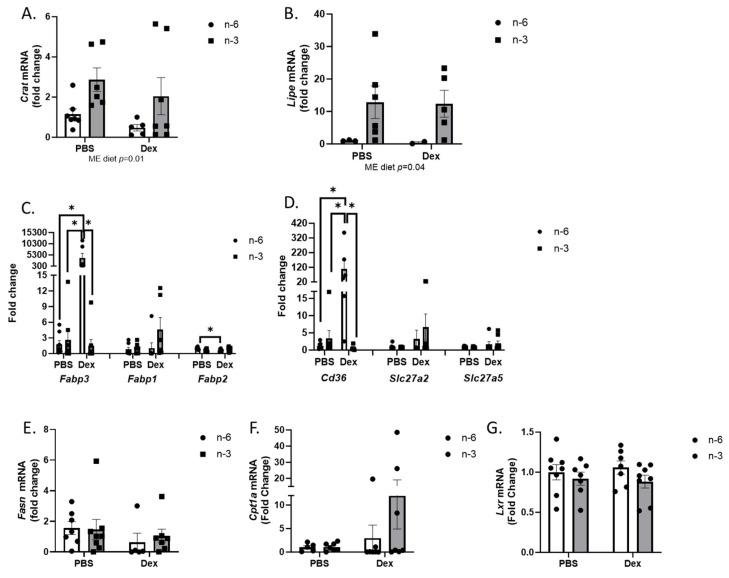
Effects of n-6 or n-3 HFD and dexamethasone on liver lipid metabolism. (**A**) Carnatine Acetyltransferase (*Crat*), (**B**) hormone sensitive lipase (*Lipe*), (**C**) fatty acid binding proteins (*Fabp3*, *Fabp1*, *Fabp2*), (**D**) fatty acid transporters (*Cd36*, *Slc27a2*, *Slc27a5*), (**E**) fatty acid synthase (*Fasn*), (**F**) carnitine palmitoyltransferase 1A (*Cpt1a*), and (**G**) liver X receptor (*Lxr)* gene expression were measured in the liver. All data are presented as mean ± SEM. Two-way ANOVA was used to analyze changes. Significance was set at *p* < 0.05. * *p* < 0.05. Main effects are denoted below each graph.

**Figure 5 ijms-24-11492-f005:**
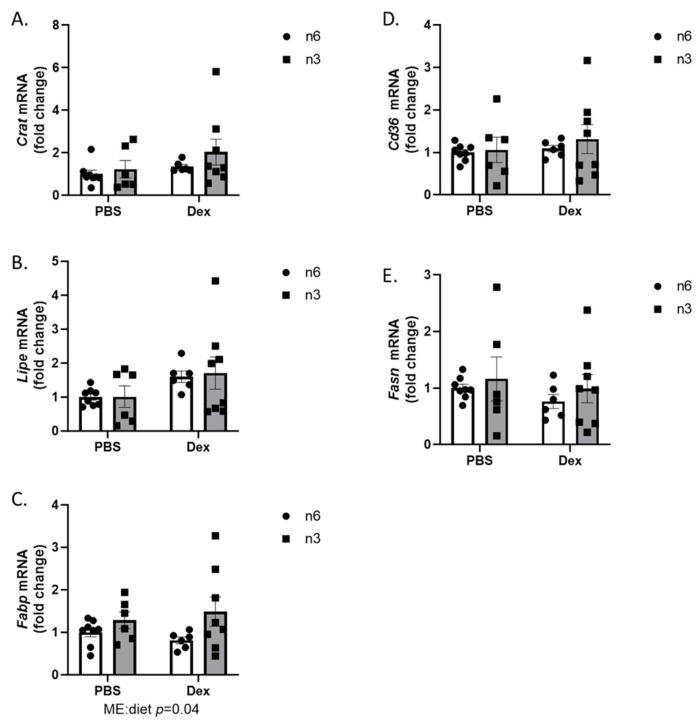
Effects of n-6 or n-3 HFD and dexamethasone on markers of lipid metabolism in glycolytic skeletal muscle. (**A**) Carnatine Acetyltransferase (*Crat*), (**B**) hormone-sensitive lipase (*Lipe*), (**C**) fatty acid-binding protein (*Fabp*), (**D**) fatty acid transporter (*Cd36*), and (**E**) fatty acid synthase (*Fasn*) gene expression were measured in the white portion of the gastrocnemius muscle. All data are presented as mean ± SEM. Two-way ANOVA was used to analyze changes. Significance was set at *p* < 0.05. Main effects are denoted below each graph.

**Figure 6 ijms-24-11492-f006:**
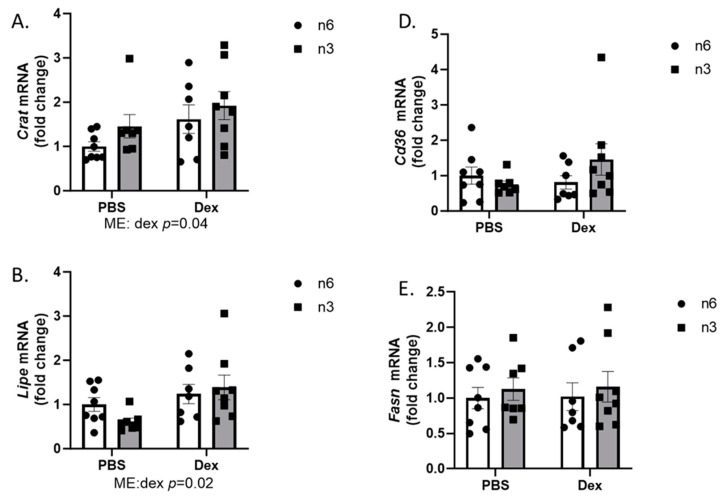
Effects of n-6 or n-3 HFD and dexamethasone on markers of lipid metabolism in oxidative skeletal muscle. (**A**) Carnatine Acetyltransferase (*Crat*), (**B**) hormone-sensitive lipase (*Lipe*), (**C**) fatty acid-binding protein (*Fabp*), (**D**) fatty acid transporter (*Cd36*), and (**E**) fatty acid synthase (*Fasn*) gene expression were measured in the red gastrocnemius muscle. All data are presented as mean ± SEM. Two-way ANOVA was used to analyze changes. Significance was set at *p* < 0.05. Main effects are denoted below each graph.

**Table 1 ijms-24-11492-t001:** Composition of high-fat lard (n-6) and high-fat Menhaden oil (n-3) diets.

Ingredient	n-6	n-3
	g	g
Lard	177.5	
Menhaden Oil, ARBP-F		177.5
Soybean Oil	25	25
Total	202.5	202.5
Saturated (%)	31.6	31.5
Monounsaturated (%)	35.5	21.8
Polyunsaturated (%)	32.9	46.7
n-6 (g)	57.0	17.9
n-3 (g)	4.4	66.6
n6/n3 ratio	13.1	0.3

**Table 2 ijms-24-11492-t002:** Gene primers for qPCR analysis.

Gene	Primers	
	Forward (5′-3′)	Reverse (5′-3′)
*Crat*	CTGTGGGATGGTGTATGAGC	CTGAGGTTCTGTTTGGCTTTC
*Lipe*	CACAGACCTCTAAATCCCACG	ATATCCGCTCTCCAGTTGAAC
*Fabp3*	GCTGGGAATAGAGTTCGACG	CTTCTCATAAGTCCGAGTGCTC
*Cd36*	GATGTGCAAAACCCAGATGAC	ACAGTGAAGGCTCAAAGATGG
*Fasn*	GATGACAGGAGATGGAAGGC	GAGTGAGGCTGGGTTGATAC
*Fabp1*	AGTCGTCAAGCTGGAAGGTGACAA	GACAATGTCGCCCAATGTCATGGT
*Fabp2*	AAAGGAGCTGATTGCTGTCCGAGA	TCGCTTGGCCTCAACTCCTTCATA
*Il-6*	GACAACTTTGGCATTGTGG	ATGCAGGGATGATGTTCTG
*Itgam*	TCCTGTACCACTCATTGTGG	GGGCAGCTTCATTCATCATG
*CD68*	CTGCTGTGGAAATGCAAGCA	TGGTCACGGTTGCAAGAGAA
*Lxr*	CCGACAGAGCTTCGTCC	CCCACAGACACTGCACAG
*Cpt1a*	GCTGGAGGTGGCTTTGGT	GCTTGGCGGATGTGGTTC
*Slc27a2*	ACACACCGCAGAAACCAAATGACC	TGCCTTCAGTGGATGCGTAGAACT
*Slc27a5*	TGTAACGTCCCTGAGCAACCAGAA	ATTCCCAGATCCGAATGGGACCAA
*Rplp0*	GCTTCATTGTGGGAGCAGACA	CATGGTGTTCTTGCCCATCAG

## Data Availability

Data sharing is not applicable to this article.

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
