# Peer review of "Effect of Omega-3 Rich High-Fat Diet on Markers of Tissue Lipid Metabolism in Glucocorticoid-Treated Mice"

_ijms, 2023, doi:10.3390/ijms241411492_

Round 1
Reviewer 1 Report
In this manuscript, Son et al. conducted research to investigate whether the replacement of n-6 PUFA with n-3 PUFA in a high-fat diet (HFD) could counteract the side effects of GC+HFD in a mouse model. The manuscript has several strengths, including the novelty of the study and the profound protective effects of n-3 in preventing liver steatosis during HFD+GC. Overall, the manuscript is well-written with minimal minor errors. However, there is still significant room for improvement, particularly in exploring the mechanisms underlying the reduction of lipid accumulation in the liver.
1. The authors mentioned changes in spleen size, which can be an indicator of systemic inflammation. Since GC can impact the inflammatory response, it would be valuable for the authors to investigate whether there are any changes in the inflammatory response in the liver or adipose tissue. This could be assessed by performing qPCR analysis of macrophage markers (e.g., Adgre1, Itgam, Ccr2, etc.) and proinflammatory cytokines (Tnf, Il1b, Ccl2, etc.).
2. In Figure 2D and 3A, the authors presented H&E staining. However, it is not indicated how large the magnification was made. This information should be provided. Additionally, in Figure 3A, it would be beneficial for the authors to show an area with portal, central, and parenchymal regions of the liver.
3. Figure 3A suggests that n-3 had no protective effects in the PBS-injected group. Is there any explanation for this observation?
4. In Figure 4, the authors measured the expression of hormone-sensitive lipase (official gene symbol: Lipe, not Hsl). However, Lipe is mainly expressed in adipocytes to regulate lipolysis and minimally expressed in hepatocytes. Considering that most fatty acids are packed as triglycerides in hepatocytes and exported as LDLs, discussing the expression of Lipe in the liver may be irrelevant.
5. In Figure 4, the authors presented the expression of Crat, indicating it as the expression of Carnitine acyltransferase in the text. However, Crat represents carnitine acetyltransferase. It is important for the authors to clarify which expression they intend to show.
6. It is known that n-3 acts as a ligand of peroxisome proliferator-activated receptor alpha (PPARα), the master regulator of fatty acid oxidation. Are there any changes in fatty acid oxidation or PPARα activity in the GC+n-3 group compared to the n-6 group? This could be assessed by measuring the expression of PPARα target genes, such as CPT1a.
7. In Figure 4, the authors indicate the expression of Fabp. However, Fabp has nine isoforms, and different Fabps are expressed in different tissues. According to the methods, Fabp3 was detected, but Fabp1 and 2 are the major Fabps in the liver.
8. The authors examined the expression of Cd36 to explain changes in fatty acid uptake. However, in addition to Cd36, Slc27 family proteins (Slc27a2 and 5) are also fatty acid transporters found in the liver, with Slc27a5 being particularly highly expressed in hepatocytes. The authors should also investigate the expression of these transporters.
9. Cd36 in hepatocytes is regulated by Liver X receptor (LXR) and PPARγ. Are there any changes in the activity or expression of these transcription factors in the GC+n-3 group compared to the n-6 group?
10. Are there any differences in serum triglyceride, cholesterol, and free fatty acids in GC+n-3 vs n-6.
11. Muscle oxidizes the fatty acid to protect lipid accumulation in the body. Are there any changes in fatty acid oxidation in the muscle?
12. Only male mice were used in the study.
13. Figure 3B, the unit is missing.
Author Response
In this manuscript, Son et al. conducted research to investigate whether the replacement of n-6 PUFA with n-3 PUFA in a high-fat diet (HFD) could counteract the side effects of GC+HFD in a mouse model. The manuscript has several strengths, including the novelty of the study and the profound protective effects of n-3 in preventing liver steatosis during HFD+GC. Overall, the manuscript is well-written with minimal minor errors. However, there is still significant room for improvement, particularly in exploring the mechanisms underlying the reduction of lipid accumulation in the liver.
We thank the reviewer for their review of our manuscript. We have tried to incorporate as many of the suggestions as we could to improve the mechanistic exploration underlying the effects that we found. We feel that the revisions have greatly enhanced the manuscript.
- The authors mentioned changes in spleen size, which can be an indicator of systemic inflammation. Since GC can impact the inflammatory response, it would be valuable for the authors to investigate whether there are any changes in the inflammatory response in the liver or adipose tissue. This could be assessed by performing qPCR analysis of macrophage markers (e.g., Adgre1, Itgam, Ccr2, etc.) and proinflammatory cytokines (Tnf, Il1b, Ccl2, etc.).
We thank the reviewer for this comment and suggestion. We have added a graph to figure 3 demonstrating no changes in IL-6 and CD68 mRNA levels, but an effect of omega-3 diet to decrease Itgam mRNA levels in liver tissue.
- In Figure 2D and 3A, the authors presented H&E staining. However, it is not indicated how large the magnification was made. This information should be provided. Additionally, in Figure 3A, it would be beneficial for the authors to show an area with portal, central, and parenchymal regions of the liver.
We have added the magnifications of the representative images in the figure legends, and changed the representative liver images to show areas with more regions of the liver.
- Figure 3A suggests that n-3 had no protective effects in the PBS-injected group. Is there any explanation for this observation?
We believe that there is no effect on n-3 in the PBS treated mice due to the short duration of the experiment. The mice were only on the respective HFDs for a total of 5 weeks from the start to end of the study and this may not be enough time to see large effects in the liver histology.
- In Figure 4, the authors measured the expression of hormone-sensitive lipase (official gene symbol: Lipe, not Hsl). However, Lipe is mainly expressed in adipocytes to regulate lipolysis and minimally expressed in hepatocytes. Considering that most fatty acids are packed as triglycerides in hepatocytes and exported as LDLs, discussing the expression of Lipe in the liver may be irrelevant.
We thank the reviewer for this comment. We have revised the manuscript to use Lipe gene symbol instead of Hsl.
- In Figure 4, the authors presented the expression of Crat, indicating it as the expression of Carnitine acyltransferase in the text. However, Crat represents carnitine acetyltransferase. It is important for the authors to clarify which expression they intend to show.
We thank the reviewer for catching this error. We measured carnitine acetyltransferase expression and have corrected this in the manuscript.
- It is known that n-3 acts as a ligand of peroxisome proliferator-activated receptor alpha (PPARα), the master regulator of fatty acid oxidation. Are there any changes in fatty acid oxidation or PPARα activity in the GC+n-3 group compared to the n-6 group? This could be assessed by measuring the expression of PPARα target genes, such as CPT1a.
We have added expression of CPT1a in the liver. There was no effects of diet or dexamethasone on CPT1a expression. These data have been added to figure 4
- In Figure 4, the authors indicate the expression of Fabp. However, Fabp has nine isoforms, and different Fabps are expressed in different tissues. According to the methods, Fabp3 was detected, but Fabp1 and 2 are the major Fabps in the liver.
We have added expression of Fabp1 and 2 in the liver. There was a decrease in n-6 fed mice treated with dexamethasone in the expression of Fabp2, and no effects of diet or dexamethasone on Fabp1 expression. These data have been added to figure 4
- The authors examined the expression of Cd36 to explain changes in fatty acid uptake. However, in addition to Cd36, Slc27 family proteins (Slc27a2 and 5) are also fatty acid transporters found in the liver, with Slc27a5 being particularly highly expressed in hepatocytes. The authors should also investigate the expression of these transporters.
We have included the expression of Slc27a2 and Slc27a5 in the liver. There were no significant differences in these markers; however there was a trend for both of them towards an increase with dexamethasone regardless of diet. These data have been added to figure 4.
- Cd36 in hepatocytes is regulated by Liver X receptor (LXR) and PPARγ. Are there any changes in the activity or expression of these transcription factors in the GC+n-3 group compared to the n-6 group?
We have added expression of LXR in the liver. There was no effects of diet or dexamethasone on LXR expression. These data have been added to figure 4
- Are there any differences in serum triglyceride, cholesterol, and free fatty acids in GC+n-3 vs n-6.
While this information would be very helpful, unfortunately we are unable to measure these markers due to lack of serum availability from this study.
Muscle oxidizes the fatty acid to protect lipid accumulation in the body. Are there any changes in fatty acid oxidation in the muscle?
We acknowledge that muscles are important metabolizer of fatty acids, which is why we included a few markers of fatty acid transport and metabolism. We were not able to measure muscle fatty acid oxidation or mitochondrial function directly in the current study although we have plans to investigate this more in the future along with some metabolomics.
- Only male mice were used in the study.
Yes, only male mice were used in this study. We acknowledge that this is a limitation to the study and that the results may differ in females. - Figure 3B, the unit is missing.
We have added units to the liver triglyceride graph.
Reviewer 2 Report
The authors of "Effect of Omega-3 Rich High-Fat Diet on Markers of Tissue Lipid Metabolism in Glucocorticoid Treated Mice" manuscript have done a commendable job. The introduction provides sufficient background and clearly defines their question. The results are thorough and experiments are well designed with sufficient number of data points. The in vivo nature of this study is significant and highly relevant. I recommend acceptance of this manuscript.
Author Response
We thank the reviewer for there kind remarks on our manuscript.
Round 2
Reviewer 1 Report
Most of my commets were properly addressed. The manuscript looks better in terms of the mechanism.